# Subjectivity of Obese Female Adolescents in South Korea Regarding Ego-Types and Characteristics

**DOI:** 10.3390/healthcare13050536

**Published:** 2025-02-28

**Authors:** Kihong Joung, Wonjae Jeon

**Affiliations:** 1Department of Physical Education, Kangnam University, 40 Gangnam-ro, Giheung-gu, Yongin-si 16979, Republic of Korea; king@kangnam.ac.kr; 2Department of Physical Education, Korea National University of Education, Gymnasium 209, Taeseongtabyeon-ro, Gangnae-myeon, Heungdeok-gu, Cheongju-si 28173, Republic of Korea

**Keywords:** obesity, female adolescents, q-methodology, physical activity, psychological well-being

## Abstract

**Background/Objectives:** The current study aims to explore the subjective ego types and characteristics of obese female adolescents in South Korea, providing insights into their self-perception and psychosocial challenges. **Methods**: Utilizing Q-methodology, a total of 49 Q-statements were derived from a comprehensive Q-population. From these, 20 adolescents diagnosed with obesity were selected as the P-Sample. Q-sorting was conducted, and the data collected were analyzed using the PQ Method software version 2.35, employing centroid factor analysis and varimax rotation. **Results**: The results revealed five distinct types, accounting for a cumulative explanatory variance of 63%. Type 1 (N = 5) was characterized as “obsession with weight loss”. Type 2 (N = 3) was identified as “overreacting to peer gaze”. Type 3 (N = 6) was labeled “lack of confidence in daily life”. Type 4 (N = 4) was described as “difficulty adapting to school life”. Finally, Type 5 (N = 2) was characterized as “passivity in physical activity”. Furthermore, the consensus statements among each type were examined in Q4 and Q7. **Conclusions**: These findings provide insights into the self-perception of obese female adolescents, emphasizing the need for tailored psychological interventions to improve their self-image and well-being.

## 1. Introduction

According to the World Health Organization (WHO), obesity is classified as a multifaceted chronic condition necessitating immediate attention. The organization’s 2022 framework underscores its prioritization as a significant public health challenge [1]. This underscores the critical need for effective prevention and management strategies, particularly among adolescents, who represent a vulnerable population group. Most importantly, the issue of youth obesity is increasingly problematic on a global scale. Adolescent obesity not only increases the size of adipocytes but augments their number, which is very likely to lead to adult obesity, and therefore requires effective public health funding and individual-level strategies to mitigate the progression to adult obesity [2].

Adolescent obesity considerably increases the probability of continuing into adulthood [3,4], underscoring the necessity for proactive intervention measures at an early stage. This progression is associated with an increased risk of metabolic disorders in adulthood, including type 2 diabetes, hypertension, and cardiovascular diseases [5]. Furthermore, experiencing stigma and social bias due to obesity in adolescence often results in severe psychological repercussions, including low self-esteem, distorted body perception, and reduced self-worth. These psychological challenges may manifest as mental health disorders, including psychological distress, anxiety, and depression, ultimately impairing social relationships [6,7]. In this context, self-esteem and peer relationships among adolescents are recognized as key psychosocial constructs that are adversely affected by obesity [6]. Self-esteem refers to an individual’s evaluation of their worth, influenced by physical attributes, social interactions, and personal achievements, and is subject to change over time [8].

In South Korea, the obesity rate among middle school students has increased by approximately 4 percentage points since 2019, reaching 19%, according to the Ministry of Education’s 2021 Sample Statistics on Student Health Checks. Additionally, about 23% of students are classified as overweight, while approximately 58% fall into the normal weight category. This indicates that one in five middle school students is classified as obese, reflecting a 2.6-fold increase in boys and a 2.2-fold increase in girls over the last decade [9]. Moreover, the prevalence of metabolic disorders among adolescents is also on the rise [10].

In female adolescents, an overabundance of adipose tissue may trigger elevated gonadotropin secretion, potentially accelerating sexual maturation [11]. This rise in sex hormones can accelerate skeletal maturation, potentially resulting in precocious puberty. Moreover, the presence of excess body fat may obscure the physical changes associated with secondary sexual characteristics, leading to delayed recognition of these changes and, in some instances, the unexpected onset of menarche [11]. Thus, the quantitative increase in the adolescent obesity population is accompanied by a serious escalation of related health issues.

While previous studies have emphasized that adolescent females exhibit lower levels of self-esteem and body satisfaction, recent research suggests that male adolescents are increasingly experiencing body image concerns, which is strongly linked to rising obesity rates and shifting societal expectations [12,13]. Recent data indicate that the prevalence of obesity among male adolescents has increased by 30% over the past decade, outpacing the rate of increase among females [12]. This shift has led to a significant rise in body dissatisfaction and self-esteem issues in both genders, highlighting the need for gender-inclusive interventions [13]. Moreover, a national survey of middle school students found that academic performance (29.6%) and physical appearance (10.9%) remain their predominant concerns, with body image issues now a growing challenge for both males and females. A gender-based analysis further reveals that, while male adolescents prioritize health, financial security, and academic success, female adolescents are more likely to express concerns about physical appearance, home environments, and peer relationships [14].

The societal emphasis on an often-unattainable ideal of thinness fosters weight-based comparisons, particularly among women, with mass media playing a significant role in perpetuating such norms [6]. As a critical developmental stage, adolescence is characterized by heightened peer influence and the pervasive impact of media [13]. Consequently, adolescents are increasingly vulnerable to becoming targets of peer bullying and may cultivate negative peer relationships, largely due to the stigmatization and discrimination associated with obesity [12]. In this context, adolescent obesity is intricately associated with various physical, mental, and social health issues that persist throughout the lifespan. Therefore, sustained attention to the challenges posed by adolescent obesity, as well as proactive efforts to develop effective interventions, is imperative [15].

Wyllie [4] observed that adolescents are particularly susceptible to internalizing negative evaluations from their peers, which can adversely affect their self-concept. During this formative period of identity development, adolescents may experience feelings of inferiority, diminished self-esteem, and impaired interpersonal relationships stemming from negative body image perceptions. These psychological ramifications can further exacerbate obesity, as evidenced by decreased athletic performance during adolescence, which may significantly influence mental health outcomes, including psychological distress, anxiety, depression, and the emergence of personality disorders throughout one’s life [16]. The findings from the studies reviewed thus far underscore the critical importance of addressing obesity among developing adolescents. Given that adolescence represents a pivotal period for physical, emotional, and social development, it is essential to conduct research on adolescent obesity from diverse perspectives, including a particular focus on female adolescents affected by obesity. In particular, obese female adolescents have a high vulnerability to body image issues and social stigma, which greatly impacts their psychosocial development [12,13]. Thus, this study aims to investigate the internal perceptions of obese female adolescents, specifically focusing on the types and characteristics of the self as perceived by this population. Furthermore, the research aims to explore the internal perceptions of obese female adolescents, focusing on their self-perception and psychosocial dynamics.

Accordingly, this study explored the perspectives of obese female adolescents by analyzing their ego-types and the defining attributes of each category. The study framework was structured around two primary research questions, shaping both the methodological approach and the data interpretation. The first inquiry examines how Q-methodology can be applied to classify the ego-types of obese female adolescents. The second question focuses on identifying the distinguishing features of each ego-type and analyzing how these traits enhance comprehension of adolescent subjectivity.

## 2. Methods

This study was conducted between 18 October 2023 and 20 April 2024, and the P-sample selection and Q-classification were conducted at two educational institutions located in a metropolitan area of South Korea. Ethical approval was obtained from the Kangnam University Institutional Review Board (IRB No. KNU-HR2109002), ensuring adherence to the principles outlined in the Declaration of Helsinki. Prior to participation, detailed information about the study’s purpose and procedures was provided to both the students and their parents/guardians. Written informed consent was obtained from all participants and their legal guardians to ensure voluntary participation.

First introduced by William Stephenson in 1953, Q-methodology serves as a research framework for investigating the individuals’ subjective viewpoints. It is particularly well-suited for analyzing intricate topics, such as personal beliefs and attitudes [17]. In contrast to R-methodology, which prioritizes researcher-defined variables, Q-methodology centers on capturing the direct expressions and viewpoints of the participants [18]. Due to its flexible approach, Q-methodology has been adopted in diverse academic fields, including psychology, healthcare studies, and educational research [19]. One distinguishing feature of Q-methodology, compared to traditional qualitative discourse analysis methods, is its structured approach to examining the participants’ responses to an identical set of Q-statements [20]. This structured methodology distinguishes itself from other qualitative techniques by enabling a more precise examination of subjective experiences. Given these strengths, Q-methodology is well-suited for assessing how obese female adolescents construct their self-identity and define their personal traits.

### 2.1. Q-Population and Q-Sample

The initial phase of Q-methodology research involves formulating a Q-population, a comprehensive collection of statements related to the study’s theme. This collection, often referred to as the ‘concourse’, serves as the foundation for participant sorting and analysis [21]. The development of these statements typically involves multiple data collection techniques, such as comprehensive literature reviews, facilitated discussions within focus groups, and structured one-on-one interviews. This method allows for the incorporation of diverse perspectives, emotional responses, and attitudinal expressions relevant to the research topic [22].

For this study, the Q-population was developed using a multi-faceted approach [23], incorporating focus group discussions with seven individuals to capture a broad spectrum of viewpoints for Q-statement formulation. This study includes a structured focus group session, comprising three students identified as obese, alongside two faculty members specializing in sports pedagogy and two experts in the sociology of sport. Each focus group discussion was held twice, with sessions lasting between one hour and one hour and twenty minutes. Following the group discussions, each student participated in a one-on-one in-depth interview to further refine their perspectives. Additionally, an extensive analysis of the literature on obesity was conducted, ultimately generating a set of 49 thematic statements for inclusion in the concourse.

Following the initial formulation, the 49 statements underwent a systematic review, where they were refined and grouped according to thematic relevance. Repetitive content was removed, while unclear statements were revised to enhance clarity and precision. To enhance the readability and precision of the statements, an external review was conducted by academic colleagues who were not directly involved in this research. Their feedback helped identify overlooked themes and refine the wording for improved clarity [24]. The finalized Q statements are presented in Table 1. To evaluate the consistency of the Q-sorting process, three respondents completed the sorting procedure twice. The results, analyzed through SPSS Statistics 22.0, produced a reliability coefficient of r = 0.69, signifying a satisfactory level of internal consistency [25].

### 2.2. P-Sample

Q-methodology is designed to capture subjective viewpoints, prioritizing individual perspectives rather than cross-participant comparisons. This methodological approach allows for flexibility in determining the number of P-sample participants [24]. A key aspect of Q-methodology is that findings derived from the P-sample do not serve as a direct representation of the general population [26].

As a technique focused on assessing personal viewpoints, Q-methodology does not impose restrictions on the number of P-sample participants. The emphasis lies in capturing individual subjectivity rather than statistical comparisons between the participants [24]. For methodological rigor, the number of P-sample participants should remain smaller than the total number of Q-statements to ensure meaningful factor analysis [27]. This study was conducted in two educational institutions—one middle school and one high school—situated in a metropolitan region of South Korea. Students classified as obese based on routine school health screenings were identified through coordination with their respective class instructors. Prior to initiating this study, necessary permissions were secured from school administrators and physical education faculty to ensure compliance with institutional guidelines. In total, 20 adolescents were chosen as the study’s P-sample. Their demographic details and factor distributions are summarized in Table 2.

### 2.3. Q-Sorting and Factor Analysis

According to Brown [28], Q-sorting is a structured approach where the participants assess a predefined set of statements. In this study, the individuals carefully examined and categorized the given Q-statements by placing them on a sorting table according to their level of agreement. The rating scale ranged from full agreement (+4) to strong disagreement (−4), allowing for a comparative analysis of subjective perspectives (see Figure 1). The Q-sorting table utilized a 9-point scale, ranging from −4 to +4.

Q-sorting methodologies can be divided into structured (forced) and unstructured (unforced) approaches. This research employed a structured sorting method, where the participants were guided to allocate a set number of statements across the predefined response categories. This ensured a balanced distribution of agreement levels, facilitating comparative analysis (see Figure 1) [29].

Between 18 March 2024 and 20 April 2024, a total of 20 participants completed the Q-sorting task following a comprehensive briefing on the study’s aims and methodology. Each participant was provided with a structured overview of the sorting process before proceeding with the exercise. The participants initially identified the statements they found most agreeable (+4) and least agreeable (−4), positioning them at opposite ends of the sorting table. The remaining statements were then systematically distributed into three categories—positive, negative, and neutral—corresponding to intermediate values ranging from +3 to −3.

To analyze the sorted data, PQ Method software (version 2.35) was employed for factor extraction. The centroid factor analysis technique was applied alongside an orthogonal varimax rotation procedure [30]. To identify the most appropriate factor solution, multiple factor models (ranging from two to seven factors) were tested, with only those demonstrating eigenvalues above 1.00 being retained for further interpretation [25].

## 3. Results

### 3.1. Eigenvalues (EVs), Variance, and Correlations

Factor analysis resulted in the extraction of five distinct types of subjectivity. The eigenvalues (EVs) for each type were recorded as 4.89, 2.48, 4.96, 4.86, and 2.18, respectively, while the variance ratios were calculated as 0.16, 0.10, 0.23, 0.15, and 0.09. The cumulative variance accounted for was 0.63, which translates to an overall explanatory power of 63% across all types. Additionally, all factors satisfied the Kaiser–Guttman criterion [31]. The results are summarized in Figure 2.

The correlation among the five types is presented in Table 3 below. A closer examination reveals that the correlation coefficients are as follows: 0.06 for Types 1 and 2, 0.09 for Types 1 and 3, 0.12 for Types 1 and 4, and 0.02 for Types 1 and 5. Additionally, the correlations for the remaining pairs are 0.12 for Types 2 and 3, 0.07 for Types 2 and 4, 0.08 for Types 2 and 5, 0.12 for Types 3 and 4, 0.20 for Types 3 and 5, and 0.15 for Types 4 and 5. The overall correlation values are predominantly low, indicating a substantial degree of explanatory power and independence among the types, thereby allowing for a clear distinction between them [26].

In examining the correlation among the ego types, the observed low correlation coefficients indicate that certain types exhibit a significant degree of independence and distinctiveness. For instance, the comparison between Type 1, characterized as “obsession with weight loss”, and Type 5, identified as “passivity in physical activity”, reveal stark differences in perceptions. Respondents in Type 1 actively pursued weight loss strategies; however, this proactive approach often culminated in an unhealthy obsession with their weight, which negatively impacted their overall well-being. By contrast, those in Type 5 are often disengaged from physical activity altogether, viewing it as burdensome rather than beneficial. This indicates that these two types not only reflect contrasting attitudes toward weight management but highlight divergent motivations for physical activity engagement.

Conversely, a relatively higher correlation was observed between Type 3, characterized by “a lack of confidence in daily life”, and Type 4, identified as “difficulty adapting to school life”. This suggests that respondents in these two types share overlapping perceptions. Specifically, Type 4 reflects the understanding that obesity adversely affects not only their self-esteem but their interactions with peers and teachers, leading to challenges in social engagement within the school environment. This shared experience indicates a relatively higher correlation between Types 3 and 4 compared to the relationship between Types 1 and 5, underscoring a notable similarity in the struggles faced by these adolescents.

### 3.2. Type 1: Obsession with Weight Loss

The participants in Type 1 showed the strongest agreement with Q1 (Z-score = 2.02), Q9 (Z-score = 1.91), Q10 (Z-score = 1.70), and Q13 (Z-score = 1.62). These statements highlight a focused and sometimes compulsive desire to lose weight, characterized by significant efforts to regulate food intake, engage in exercise, and adhere to dieting routines. The high Z-score for Q1, emphasizing frustration with specific areas of the body resistant to weight loss, reflects the emotional toll of striving for an ideal body image.

Conversely, Q5 (Z-score = −2.04), which deals with a lack of confidence in daily life due to body shape, was the most negatively rated statement. This indicates that, while the participants in this type are intensely preoccupied with weight control, their concerns do not necessarily diminish their self-confidence in everyday activities.

This type comprised five participants: P1 (0.66), P7 (0.58), P11 (0.91), P12 (0.79), and P17 (0.89). Notably, P11 had the highest factor weight, strongly embodying the defining characteristics of this group. The findings suggest that societal pressures and personal aspirations for thinness drive these individuals, potentially fostering both motivation and psychological distress. The characteristics of this type are summarized in Table 4, which presents the key statements and factor weights associated with this category.

### 3.3. Type 2: Overreacting to the Peer Gaze

For Type 2, the participants resonated most strongly with Q2 (Z-score = 1.92), Q3 (Z-score = 1.80), and Q8 (Z-score = 1.65). These statements underscore heightened sensitivity to others’ perceptions, particularly around being observed or judged in social contexts. The central theme for this type revolves around external validation and the discomfort arising from peer scrutiny.

By contrast, Q11 (Z-score = −1.95), which involves managing weight through specific dietary habits, was rated most negatively. This suggests that, while the participants in this group are acutely aware of external judgment, they may prioritize emotional responses to social interactions over internalized weight-management behaviors.

This group consisted of three participants: P2 (0.75), P8 (0.83), and P13 (0.69). Among them, P8 exhibited the highest factor weight, making their responses particularly illustrative of the group’s defining traits. This type reflects the interplay between social comparison and self-consciousness, themes commonly observed in adolescence. Table 4 provides a summary of the main statements and factor loadings for this type, emphasizing the heightened sensitivity to peer perceptions.

### 3.4. Type 3: A Lack of Confidence in Daily Life

The participants classified under Type 3 strongly agreed with Q4 (Z-score = 2.15), Q5 (Z-score = 1.95), and Q18 (Z-score = 1.71). These statements reflect a pervasive struggle with self-esteem and confidence, particularly influenced by their physical appearance. Q4, which highlights sadness when comparing oneself to slimmer individuals, underscores a central theme of self-perception shaped by societal standards.

Negative responses were strongest for Q14 (Z-score = −1.93), Q22 (Z-score = −1.64), and Q23 (Z-score = −1.62), which pertain to explicit social barriers, like peer exclusion or difficulties engaging in weight-loss activities. This indicates that the participants in this group internalize their insecurities, rather than attributing them to external social dynamics.

This type included six participants: P3 (0.89), P4 (0.91), P9 (0.70), P14 (0.69), P18 (0.84), and P19 (0.76). With the highest factor weight, P4 serves as the most representative individual for this type. The findings suggest that interventions targeting self-esteem and body positivity may significantly benefit this group. The defining characteristics of this type, as presented in Table 4, illustrate the impact of low self-esteem and body dissatisfaction on daily activities.

### 3.5. Type 4: Difficulty Adapting to School Life

The participants in Type 4 identified Q14 (Z-score = 2.02), Q16 (Z-score = 1.78), and Q24 (Z-score = 1.67) as the most positive statements, reflecting the challenges of navigating peer relationships and academic pressures within the school environment. Q14, which emphasizes feeling ignored or teased, was particularly salient, indicating the emotional and social difficulties these adolescents experience.

On the other hand, Q7 (Z-score = −1.92), which relates to dissatisfaction with one’s reflection, received the lowest agreement. This suggests that the participants’ challenges are more rooted in external social factors than internalized body image concerns.

This group comprised four participants: P5 (0.63), P6 (0.66), P15 (0.61), and P16 (0.87). Among these, P16’s responses stood out with the highest factor weight, providing a clear representation of this type. The results emphasize the importance of school-based interventions that foster inclusivity and peer support. As detailed in Table 4, the statements and factor loadings for this type reveal challenges related to social interactions and school adaptation

### 3.6. Type 5: Passivity in Physical Activity

Type 5 participants expressed strong agreement with Q15 (Z-score = 1.98), Q19 (Z-score = 1.83), and Q21 (Z-score = 1.60). These statements reflect a disengaged attitude toward physical activity, often viewing it as a burden rather than an opportunity for health or enjoyment.

By contrast, Q8 (Z-score = −2.02), Q16 (Z-score = −1.92), and Q25 (Z-score = −1.79) were the most negatively rated, highlighting a lack of concern about peer judgment or teasing in relation to their inactivity. This suggests that their disengagement is less about social anxiety and more about an intrinsic lack of motivation.

This type was represented by two participants: P10 (0.81) and P20 (0.73). With the highest factor weight, P10 exemplifies this type’s passive stance toward physical activity. These findings suggest that interventions focused on intrinsic motivation and fostering a sense of accomplishment in physical activity could benefit this group. Table 4 summarizes the core characteristics of this type.

### 3.7. Consensus Statements

The commonly agreed-upon statements for all types were Q4 and Q7, with Z-scores of 0.82 and 0.75, respectively, highlighting areas of consensus across the identified ego types.

## 4. Discussion

The current investigation explored the eigenvalues (EVs), variance, and interrelationships among the five identified ego types of obese female adolescents. The classification outcomes for the P-Sample and Q-Sample corresponding to each type were scrutinized using Z-scores. The additional discussion for each type based on the results of this study is as follows.

The findings on Type 1 (‘obsession with weight loss’) emphasize the strong influence of societal standards and media portrayals on adolescent self-perception [12,13,32]. Unrealistic ideals of thinness, perpetuated by media, often lead adolescents to engage in harmful behaviors, such as irregular eating patterns and extreme dieting. These behaviors not only result in physical health risks, including anemia and undernutrition [33,34], but exacerbate psychological challenges, such as anxiety and depression. Addressing these societal pressures is critical to reducing the harmful fixation on weight loss among adolescents and promoting healthier self-perceptions.

Research by Becker et al. [35] emphasizes the potential of peer-led cognitive dissonance-based intervention programs. These initiatives aim to address the contradictions inherent in society’s preference for unrealistic thinness, promoting positive body image among adolescents through small group, peer-led activities. These programs underscore the importance of altering entrenched beliefs and behaviors related to body image.

Adolescence is marked by a heightened tendency toward self-objectification, wherein individuals continuously assess their appearance against socially endorsed standards. This phenomenon is particularly pronounced among female adolescents compared to their male counterparts. Notably, female adolescents report significantly lower levels of body satisfaction, a disparity that becomes increasingly evident during adolescence [12,13,16]. Furthermore, the compulsive drive for weight loss significantly increases the risk of disordered eating behaviors among female adolescents. Many engage in extreme dieting followed by binge eating, which may lead to the abuse of purging methods, such as vomiting or the misuse of laxatives, ultimately resulting in conditions like anorexia or bulimia [36,37]. Evidence suggests that overweight and moderately obese adolescents are at a higher risk of developing eating disorders compared to their underweight peers. Specifically, 9.7% of underweight adolescents, 50.5% of overweight adolescents, and 38.5% of those classified as moderately obese fall within the high-risk category for eating disorders [38]. This alarming trend is underscored by research indicating that the incidence of eating disorders among female adolescents in the United States has surged more than eightfold over the past decade [39].

In light of these findings, it is crucial to address the psychosocial factors contributing to the unhealthy fixation on weight loss among female adolescents. By fostering a supportive environment that prioritizes healthy body image and encourages positive self-perception, targeted interventions can mitigate the risks associated with disordered eating and promote overall well-being.

The findings related to Type 2, which emphasize the theme of “overreacting to the peer gaze”, align closely with Festinger’s (1954) concept of social comparison [40]. This theory posits that individuals evaluate themselves by comparing their actions, attributes, and achievements to those of their peers. Adolescence is a critical period during which such social comparisons are particularly salient, especially concerning physical appearance, sexual maturation, and familial background. During this time, adolescents are acutely aware of how their outward appearance is perceived by others, significantly influencing their self-assessment [12].

Dissatisfaction with body image is often cited as a primary barrier to physical activity among obese adolescents [41,42]. Research indicates that obese female students exhibit lower body image scores compared to their peers of normal weight, and this dissatisfaction substantially affects their participation in physical activities [43]. Furthermore, the health concerns associated with obesity can become distorted into issues of appearance, leading to psychological problems such as intense feelings of inferiority, body image dissatisfaction, and a weakened self-concept [6,42], In severe cases, this can culminate in serious mental health issues, including the need for intervention for anorexia and bulimia [44]. These studies highlight that obese female adolescents often perceive their physical issues as emotionally and socially problematic.

Obese female students frequently experience significant anxiety about exposing their unsatisfactory body shapes to others, particularly during physical education classes. The pressure to reveal their bodies is amplified in situations where attention is focused on them, such as during group activities or when wearing revealing athletic attire in coeducational settings. The presence of male peers often intensifies this discomfort. While these students may attempt to conceal their bodies, the nature of physical activity makes it challenging to do so effectively. Consequently, they may internalize societal perceptions that equate their obesity with a lack of willpower, greed, or laziness, a narrative that reinforces negative self-judgment [45]. This internalized stigma can lead them to avoid engaging fully in physical activities, further compounding their difficulties.

Contemporary culture has embraced an ideology that idealizes thinness, often reinforcing social hierarchies based on body size [46]. Within this framework, health is often defined in terms of slimness, reinforcing a surveillance system that scrutinizes appearances and positions fat bodies as unhealthy or even harmful [47]. This perspective places the responsibility for obesity on the individual, framing it as a result of personal failings, such as laziness or gluttony. Such attitudes cultivate an environment where public condemnation of obesity is normalized [48]. As a result, individuals with obesity may grapple not only with physical weight but with the psychological burden associated with societal judgment.

The discussion regarding Type 3 begins with the exploration of self-objectification theory, which is particularly relevant to understanding the psychological landscape of female adolescents. The cultural and social objectification of women’s bodies results in a significant number experiencing body dissatisfaction and lowered self-esteem, thereby giving rise to various psychological issues [49]. In contemporary society, women are bombarded with images of the slender body as the ideal, propagated through mass media. By contrast, men’s ideal body image is characterized by a muscular physique that falls within the normal range of the body mass index [50]. Furthermore, research indicates that women are more dissatisfied with their bodies and react more strongly to media-constructed body images, whereas men tend to prioritize social achievements and athletic performance over appearance [49,51].

Recent studies highlight that obesity-related psychological distress is not exclusive to older adults but is increasingly prevalent among adolescents. Obese adolescents report higher levels of body dissatisfaction, anxiety, and depressive symptoms, which persist into adulthood and significantly impact mental well-being [52,53]. Participation in physical activity has been shown to counteract obesity-related cognitive decline, with research highlighting its notable impact on adolescents experiencing severe obesity and metabolic complications [54]. Thus, physical activity emerges as a critical intervention for preventing and treating the physical, emotional, social, and cognitive challenges faced by obese adolescents.

However, obese female students often demonstrate a lack of intrinsic motivation to engage in physical activity. This deficiency may lead to a decreased willingness to participate, especially when confronted with uncomfortable or intimidating physical environments during physical education classes. To foster a more proactive approach to participation, it is essential to enhance their perceived physical self-efficacy, which is defined as the self-assessment of one’s physical capabilities in achieving set goals related to physical activity [55]. Practical strategies for achieving this include the following: 1. Peer-Led Physical Activity Programs: Adolescents can collaborate with peers in structured physical activity programs that promote social support and build confidence in their abilities [56]. 2. Goal-Oriented Exercise Plans: Tailored exercise plans with achievable milestones can gradually improve adolescents’ physical competence and reinforce their sense of accomplishment [55,56]. 3. Positive Reinforcement: Providing verbal encouragement and celebrating progress can enhance motivation and focus on personal growth rather than perceived physical limitations [57]. 4. Role Models and Success Stories: Exposure to relatable role models or peer success stories can inspire adolescents to engage in physical activities and view them as achievable and beneficial [56]. 5. Inclusive School-Based Programs: Schools can implement non-competitive, inclusive physical activities that create a supportive environment, reducing the stigma often associated with obesity in physical education classes [55,56].

Regarding Type 4, the discussion is as follows. Individuals who are obese are often perceived as lazy, greedy, lacking self-control, and physically unattractive [48]. The stigma associated with obesity explains how these individuals are often targeted by violence and discrimination from peers and family, leading to negative interpersonal relationships [58,59]. Adolescents spend a significant amount of time in school, where peers serve as sources of emotional support and connection, providing the practical assistance necessary for adapting to their surroundings while modeling various social behaviors [60,61]. Those adolescents who receive acceptance from their peers and establish stable relationships tend to exhibit higher levels of self-esteem. Conversely, adolescents who face rejection from their peers and fail to form positive relationships often engage in negative self-assessment [60]. Notably, female adolescents are more likely to depend on their peers and respond sensitively to peer evaluations, which results in a more pronounced impact of peer relationships on their self-esteem compared to their male counterparts [62].

Reiter-Purtill et al. [63] demonstrated that peer victimization among adolescents with severe obesity undermines their self-worth, thereby influencing their psychosocial maladjustment. The finding that obesity impacts the interpersonal competence of female adolescents, fully mediated by peer relationships, aligns with previous studies that identified the mediating effects of peer victimization or negative peer relationships on the association between obesity and social competence constructs adopted in this study, such as social effectiveness and psychosocial adaptation [41,64].

In the context of Type 5, the self-image of obese female adolescents who have naturally distanced themselves from physical activity was identified. The prevalence of adolescent obesity is increasing at a concerning rate, with projections suggesting that, by 2030, nearly 50% of the global adult population will be classified as either overweight or obese [65]. This escalating trend underscores the urgent need for comprehensive intervention strategies. Obesity is also closely related to changes in the autonomic nervous system, which regulates vital functions necessary for sustaining life, including respiration, digestion, circulation, absorption, secretion, and reproduction [66]. However, imbalances in the autonomic nervous system can lead to various diseases, making individuals particularly susceptible to cardiovascular conditions [67].

The participants in this study likewise demonstrated a perception that their obesity limited their participation in physical activities. Constructive measures to address these issues could center on promoting physical activity. Engaging in regular physical activity is an effective strategy for mitigating the problems associated with obesity during adolescence. Specifically, physical activity has proven effective in both the prevention and treatment of lifestyle-related diseases such as obesity, diabetes, and cardiovascular conditions, contributing to improved physical health and overall well-being [14,68].

In conclusion, adolescent obesity is intertwined with physical, mental, and social health issues throughout the lifespan. In addition to challenges related to school and physical activity, the participants reported significant influences from their family and social environments. Supportive family dynamics and positive peer interactions were noted as critical factors in shaping their self-perceptions and engagement in physical activities. Therefore, there is an urgent need for sustained attention to the problem of adolescent obesity from schools, families, and media, alongside the implementation of educational alternatives by governmental bodies and educational departments.

## 5. Conclusions

This study explored the subjective self-perception types and characteristics of obese female adolescents in South Korea using Q-methodology. Five distinct ego types were identified: Type 1, characterized by an obsession with weight loss; Type 2, marked by overreacting to peer scrutiny; Type 3, defined by a lack of confidence in daily life; Type 4, described as difficulties adapting to school life; and Type 5, characterized by passivity in physical activity. These findings underscore the diverse challenges faced by obese female adolescents and highlight the need for targeted interventions tailored to each ego type.

The results emphasize the importance of early interventions that address psychological, social, and physical barriers to promote positive self-perception and physical activity engagement. For instance, strategies such as peer-led physical activity programs, personalized goal-setting, and inclusive school-based activities could foster a supportive environment that enhances self-efficacy and mitigates the stigma associated with obesity.

Future research should investigate the long-term evolution of these ego types and evaluate the effectiveness of tailored interventions in improving both physical and psychological outcomes for obese adolescents. Cross-cultural studies may also provide valuable insights into how societal and cultural factors influence self-perception and obesity-related behaviors.

By identifying and addressing the subjective challenges of obese female adolescents, this study offers a foundation for developing comprehensive and practical interventions aimed at improving their well-being and quality of life.

## 6. Limitations of the Study

This study was geographically restricted to South Korea, and cultural factors may have influenced the participants’ responses. The complexity of obesity as a global health issue necessitates further investigation in diverse cultural contexts to ensure the generalizability of findings [9,27]. Additionally, addressing the interplay between societal norms, media influence, and individual experiences could enrich our understanding of the disorder’s psychosocial and physical dimensions. South Korea’s societal norms regarding body image and obesity are shaped by unique cultural pressures, such as high media influence and peer scrutiny. Consequently, the findings may not fully capture the experiences of obese female adolescents in other cultural contexts. Cross-cultural studies could provide valuable comparative insights and help determine the universality or cultural specificity of the identified ego-types.

Additionally, while this study focused on understanding subjective experiences, it relied on self-reported data collected through Q-sorting. This approach may have been influenced by social desirability bias, where the participants provide responses they believe are more socially acceptable rather than fully reflecting their true perspectives. Combining Q-methodology with other qualitative or quantitative methods, such as in-depth interviews or surveys, could enhance the robustness of the findings.

Lastly, this study did not longitudinally examine how the identified ego-types and characteristics evolve over time or in response to interventions. A longitudinal design would provide valuable insights into the stability of these ego-types and their potential role in shaping psychosocial development and well-being. Future research could explore these dynamics to inform more effective and targeted intervention strategies.

## Figures and Tables

**Figure 1 healthcare-13-00536-f001:**
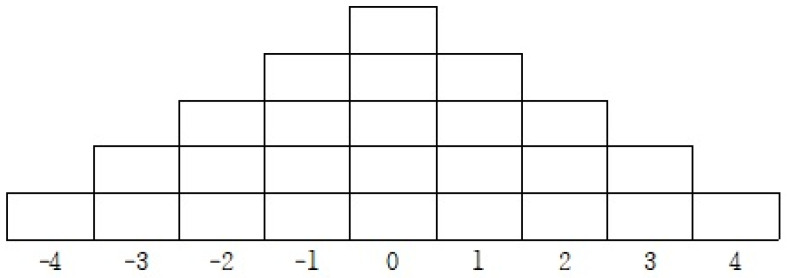
Q sorting response table.

**Figure 2 healthcare-13-00536-f002:**
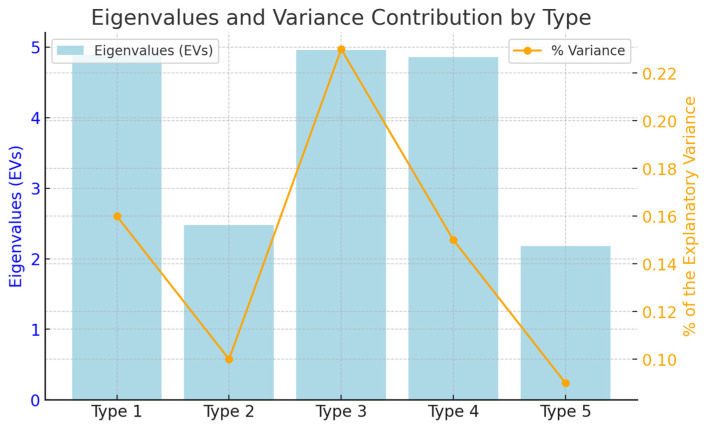
Eigenvalues (EVs) and variance between types.

**Table 1 healthcare-13-00536-t001:** Q statements.

Q Number	Q Statements
1	I feel very frustrated because certain areas of my body do not lose weight despite dieting.
2	I worry that others will notice the imperfections on my thighs and buttocks.
3	Thinking about how others perceive my appearance makes me feel depressed.
4	Comparing myself to slender individuals makes me feel sad.
5	I lack confidence in my daily life due to my body shape.
6	Concentrating on other tasks is difficult when I think about my physical appearance.
7	I feel dissatisfied when I see my reflection in mirrors or windows.
8	I sense that people around me view my body negatively.
9	I constantly need to regulate my food intake.
10	I must make every effort to reduce my body weight.
11	I must consume specific foods to control my weight.
12	I feel that I must engage in exercise to achieve weight loss.
13	I do not receive social support from my family.
14	Friends sometimes ignore or tease me.
15	I become easily out of breath due to my weight.
16	I feel embarrassed and burdened during conversations with teachers.
17	Overall, I feel like a failure.
18	I believe I have little to be proud of regarding myself.
19	I frequently receive unsolicited advice about participating in exercise from those around me.
20	Occasionally, I feel like I am a worthless person.
21	I often feel excluded from sports activities.
22	I find it difficult to participate in class actively.
23	I find it challenging to engage in various exercises aimed at weight loss.
24	I struggle to connect easily with my peers at school.
25	I feel that my classmates are bullying me at school.

**Table 2 healthcare-13-00536-t002:** Summary of characteristics for the P-sample and Factor weights.

Type (Factor)	Ego-Type Description	P-Sample	Grade	Affiliation (School)	Regular Engagement in Physical Activity	Factor Weight
Type 1. (N = 5)	Obsession with weight loss	1	1st	Middle	Once a week	0.66
		7	3rd	High	Absence	0.58
		11	1st	High	Absence	0.91
		12	3rd	Middle	Once a month	0.79
		17	3rd	High	Absence	0.89
Type 2. (N = 3)	Overreacting to peer gaze	2	3rd	Middle	Absence	0.75
		8	1st	High	Biweekly	0.83
		13	2nd	Middle	Once a week	0.69
Type 3. (N = 6)	A lack of confidence in daily life	3	3rd	High	Once a month	0.89
		4	3rd	Middle	Biweekly	0.91
		9	3rd	Middle	Absence	0.70
		14	2nd	Middle	Absence	0.69
		18	1st	High	Once a month	0.84
		19	2nd	High	Once a week	0.76
Type 4. (N = 4)	Difficulty adapting to school life	5	3rd	High	Once a month	0.63
		6	2nd	Middle	Absence	0.66
		15	2nd	Middle	Once a week	0.61
		16	2nd	High	Absence	0.87
Type 5. (N = 2)	Passivity in physical activity	10	2nd	High	Absence	0.81
		20	3rd	Middle	Absence	0.73

**Table 3 healthcare-13-00536-t003:** Correlations between types.

	Type 1	Type 2	Type 3	Type 4	Type 5
Type 1	1				
Type 2	0.06	1			
Type 3	0.09	0.12	1		
Type 4	0.12	0.07	0.12	1	
Type 5	0.02	0.08	0.20	0.15	1

**Table 4 healthcare-13-00536-t004:** Statements with Z-scores of ±1.00 (or higher) from Type 1 to Type 5.

		Q-Statement Number	Z-Score
Type 1	Positive	1	2.02
		9	1.91
		10	1.70
		13	1.62
	Negative	5	−2.04
		7	−1.81
Type 2	Positive	2	1.92
		3	1.80
		8	1.65
	Negative	11	−1.95
		12	−1.82
		20	−1.79
Type 3	Positive	4	2.15
		5	1.95
		18	1.71
	Negative	14	−1.93
		22	−1.64
		23	−1.62
Type 4	Positive	14	2.02
		16	1.78
		24	1.67
		25	1.62
	Negative	7	−1.92
		10	−1.89
		13	−1.76
Type 5	Positive	15	1.98
		19	1.83
		21	1.60
		23	1.59
	Negative	8	−2.02
		16	−1.92
		25	−1.79

## Data Availability

The data presented in this study are available upon request from the corresponding authors.

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
