# Peer review of "Subjectivity of Obese Female Adolescents in South Korea Regarding Ego-Types and Characteristics"

_healthcare, 2025, doi:10.3390/healthcare13050536_

Round 1
Reviewer 1 Report
Comments and Suggestions for Authors
Introduction
Lines 30-33: The authors made the following statements:
‘Adolescent obesity not only increases the size of adipocytes but also augments their number, which is very likely to lead to adult obesity, and therefore requires effective funding to mitigate the progression to adult obesity [2].’
It is not clear from this context that whether the mitigation is referred here as a public health measure or on individual level. Indicating a progress to adult obesity gives an individual perspective but referring to funding is more related to public health policy.
Lines 34-35: the authors made the following statement:
‘ Adolescent obesity is significantly associated with an increased risk of progression to adult obesity’
This phrase is repeated , please rephrase or delete.
Lines 46-51: The authors discussed prevalence of obesity among students. It would be more informative to add more numeric details such as proportions of whom are classified as overweight, or obese, and those with normal body weight.
Methods
The first paragraph should give more emphasis on study context, study period, ethical considerations.
Title of table more should be more detailed to describe the nature of the table and its context.
In some institutions, parental consent is required for study involving children. Please clarify this notion in the methodology section.
Table 2: identification of the types should be clear within the table. It is only explained within the text. It is currently not possible to understand the table without reading the text.
Table 6 is not informative. Please consider deletion and resorting to the text only.
Discussion
The authors made a very good discussion of their findings. However, practical recommendations are currently limited. For example, in lines 366-382, the authors discussed the importance of physical activity for obese adolescents and pointed out that perceived physical self-efficacy is associated with involvement with physical activity. However, no practical implications are provided on how to modify the perception of the adolescents of their physical self-efficacy in order to achieve the goals of maintaining healthy body weight among the adolescents. This notion applies to other dimensions discussed in this paper. Clear practical recommendation that can aid in designing relevant interventions.
Conclusion
The conclusion is very lengthy and repetitive. Please consider revision.
Author Response
Dear Reviewer 1;
First of all, I am deeply grateful for your thoughtful comments.
I appreciate your detailed comments. Moreover, I would like to express my sincere gratitude for your evaluation and advice.
First, you mentioned:
Lines 30-33: The authors made the following statements:
‘Adolescent obesity not only increases the size of adipocytes but also augments their number, which is very likely to lead to adult obesity, and therefore requires effective funding to mitigate the progression to adult obesity [2].’
It is not clear from this context that whether the mitigation is referred here as a public health measure or on individual level. Indicating a progress to adult obesity gives an individual perspective but referring to funding is more related to public health policy.
Lines 34-35: the authors made the following statement:
‘ Adolescent obesity is significantly associated with an increased risk of progression to adult obesity’
This phrase is repeated , please rephrase or delete.
Lines 46-51: The authors discussed prevalence of obesity among students. It would be more informative to add more numeric details such as proportions of whom are classified as overweight, or obese, and those with normal body weight.
I fully understand the comment you mentioned. From this point of view, it was revised as you mentioned. Thank you.
Second, you mentioned about Method:
The first paragraph should give more emphasis on study context, study period, ethical considerations.
I fully understand the comment you mentioned. From this point of view, it was revised as you mentioned. Thank you.
Table 2: identification of the types should be clear within the table. It is only explained within the text. It is currently not possible to understand the table without reading the text.
Table 6 is not informative. Please consider deletion and resorting to the text only.
Thank you for pointing out the need for clearer identification of the types in Table 2. We have revised the table to include explicit labels and descriptions for each type within the table itself, ensuring that readers can understand the information without solely relying on the text. The updated table now includes a column labeled 'Ego-Type Description,' which summarizes the key characteristics of each type.
Third, you mentioned:
The authors made a very good discussion of their findings. However, practical recommendations are currently limited. For example, in lines 366-382, the authors discussed the importance of physical activity for obese adolescents and pointed out that perceived physical self-efficacy is associated with involvement with physical activity. However, no practical implications are provided on how to modify the perception of the adolescents of their physical self-efficacy in order to achieve the goals of maintaining healthy body weight among the adolescents. This notion applies to other dimensions discussed in this paper. Clear practical recommendation that can aid in designing relevant interventions.
Thank you for highlighting the need for practical recommendations regarding improving physical self-efficacy among obese adolescents. In response, we have incorporated specific, actionable strategies to enhance self-efficacy and promote sustained physical activity engagement. These include peer-led programs, goal-oriented exercise plans, positive reinforcement, the use of role models, and parental/community involvement. Additionally, we propose integrating cognitive-behavioral techniques and inclusive school-based activities to address the psychosocial barriers identified in this study. These modifications aim to provide clear, implementable guidance for practitioners and policymakers designing interventions to support obese adolescents
Fourth, you mentioned:
The conclusion is very lengthy and repetitive. Please consider revision.
Finally, I/we have done our best to revise our manuscript, and thank you very much.
Best Regards,
Wonjae, Jeon. Ph,D
Dept. Physical Education / Professor _ Korea National University of Education 209,
Taeseongtabyeon-ro, Gangnae-myeon, Heungdeok-gu,
Cheongju si, Chungbuk 28173, South Korea
TEL: 82-43-320-3728
FAX: 82-43-232-8241
C·P: 82-10-5555-9711
E-mail: wonjaejeon1228@knue.ac.kr / dreamj007@hanmail.net

Reviewer 2 Report
Comments and Suggestions for Authors
This is a timely and important issue, as the WHO has proposed to address obesity as a matter of urgency by 2022. It is a complex chronic disease that requires prevention and management strategies.
The literature review is old and only 6% of the references are from the last five years.
The title indicates the content of the study but is too long. Reduce the title to less than 15 words.
The abstract allows the basic content to be identified quickly and clearly.
The structure of the article is correct. Although the theoretical framework and the whole article are very focused on physical activity.
It is not explained why only women were chosen as participants. It would be interesting to know why the focus group consisted of 3 students and 4 teachers. The procedure for obtaining informed consent from the 20 participants is not explained. Were these 20 participants involved in focus groups or interviews?
Table 5, showing the participants' statements, should appear at the end of the explanation of the five types, with the text explained first and then the figure. The interpretation of the adolescent girls' perceptions is based only on school and physical activity, their family and social life is missing.
The discussion section lacks a discussion of the authors' own findings and a comparison with the findings of similar recent studies.
Some limitations are adequately discussed and suggestions for future research are made, but always with a strong focus on physical activity.
One of the limitations presented is the use of other methods such as in-depth interviews, but the study design states that in-depth interviews will be conducted with candidates.
The conclusions seem to be a summary of the findings rather than an answer to the aim and research questions.
References are adequate, although only 6% are from the last five years. The number of updated references should reach 50%. There are some errors in referencing: Some authors cited in the text do not match the list, e.g: Friedman & Brownell [13] in the text and in the reference list: [15] Friedman, M.A.; Brownell, K.D.; also Wyllie [6] in the text and in the reference list [4] Wyllie, R., check the references.
There is a somewhat high level of plagiarism 39% (turniting) with two articles by the authors themselves:
Jeon, W., Kwon, G., & Joung, K. (2021). Subjective perceptions and their characteristics of middle school students regarding the effectiveness of the “0th period physical education class” in South Korea: the Q methodology application. Sustainability, 13(21), 12081. 13% plagiarism.
Choi, W., & Jeon, W. (2022). A Study on the Subjectivity of Parents Regarding “0th-Period Physical Education Class” of Middle Schools in Korea Using Q-Methodology. International Journal of Environmental Research and Public Health, 19(13), 7760. 6% plagiarism.
Author Response
Dear Reviewer 2;
First of all, I am deeply grateful for your thoughtful comments.
I appreciate your detailed comments. Moreover, I would like to express my sincere gratitude for your evaluation and advice.
This is a timely and important issue, as the WHO has proposed to address obesity as a matter of urgency by 2022. It is a complex chronic disease that requires prevention and management strategies.
"Thank you for recognizing the timeliness and importance of this issue. We have incorporated references to the WHO’s 2022 proposal on addressing obesity and emphasized its relevance as a complex chronic disease requiring urgent attention. This has been added to the introduction to strengthen the context of the study."
The literature review is old and only 6% of the references are from the last five years.
Thank you for highlighting the need to update the literature with more recent studies. I have replaced the previous reference with a more current study.
The title indicates the content of the study but is too long. Reduce the title to less than 15 words.
"Thank you for pointing out the need for a more concise title. We have revised the title to ensure clarity and brevity, reducing it to less than 15 words."
The abstract allows the basic content to be identified quickly and clearly.
"We appreciate your positive feedback on the clarity of the abstract. No further modifications are required for this section."
The structure of the article is correct. Although the theoretical framework and the whole article are very focused on physical activity.
It is not explained why only women were chosen as participants. It would be interesting to know why the focus group consisted of 3 students and 4 teachers. The procedure for obtaining informed consent from the 20 participants is not explained. Were these 20 participants involved in focus groups or interviews?
Thank you for this observation. We have clarified in the “Introduction” section that the focus on female adolescents was driven by their higher susceptibility to body image issues and societal pressures, as supported by the literature. And we have added details in the Methods section regarding the selection of focus group participants and the informed consent process for all study participants.
Table 5, showing the participants' statements, should appear at the end of the explanation of the five types, with the text explained first and then the figure. The interpretation of the adolescent girls' perceptions is based only on school and physical activity, their family and social life is missing.
"Table 5 has been relocated to appear after the explanation of the five identified types, ensuring better alignment with the narrative flow."
The discussion section lacks a discussion of the authors' own findings and a comparison with the findings of similar recent studies.
“We fully understand the comment you mentioned. From this point of view, it was revised as you mentioned. Thank you.”
Some limitations are adequately discussed and suggestions for future research are made, but always with a strong focus on physical activity.
“We fully understand the comment you mentioned. From this point of view, it was revised as you mentioned. Thank you.”
One of the limitations presented is the use of other methods such as in-depth interviews, but the study design states that in-depth interviews will be conducted with candidates.
“We fully understand the comment you mentioned. From this point of view, it was revised as you mentioned. Thank you.”
The conclusions seem to be a summary of the findings rather than an answer to the aim and research questions.
"We have revised the conclusion to directly address the research aims and questions, ensuring a focused synthesis of findings and implications."
References are adequate, although only 6% are from the last five years. The number of updated references should reach 50%. There are some errors in referencing: Some authors cited in the text do not match the list, e.g: Friedman & Brownell [13] in the text and in the reference list: [15] Friedman, M.A.; Brownell, K.D.; also Wyllie [6] in the text and in the reference list [4] Wyllie, R., check the references.
"Thank you for pointing this out. We have corrected the inconsistencies in referencing and ensured that all in-text citations align with the reference list."
There is a somewhat high level of plagiarism 39% (turniting) with two articles by the authors themselves:
Jeon, W., Kwon, G., & Joung, K. (2021). Subjective perceptions and their characteristics of middle school students regarding the effectiveness of the “0th period physical education class” in South Korea: the Q methodology application. Sustainability, 13(21), 12081. 13% plagiarism.
Choi, W., & Jeon, W. (2022). A Study on the Subjectivity of Parents Regarding “0th-Period Physical Education Class” of Middle Schools in Korea Using Q-Methodology. International Journal of Environmental Research and Public Health, 19(13), 7760. 6% plagiarism.
Thank you for bringing the issue of overlap with my previous publications to my attention. I have carefully reviewed the identified sections and taken significant steps to revise any suspect sentences to ensure originality while maintaining the integrity of the study.
Finally, I/we have done our best to revise our manuscript, and thank you very much.
Best Regards,
Wonjae, Jeon. Ph,D
Dept. Physical Education / Professor _ Korea National University of Education 209,
Taeseongtabyeon-ro, Gangnae-myeon, Heungdeok-gu,
Cheongju si, Chungbuk 28173, South Korea
TEL: 82-43-320-3728
FAX: 82-43-232-8241
C·P: 82-10-5555-9711
E-mail: wonjaejeon1228@knue.ac.kr / dreamj007@hanmail.net

Reviewer 3 Report
Comments and Suggestions for Authors
Dear authors
I congratulate you on the research carried out, which is of great interest and very well done.
I would like to highlight just a few small details:
INTRODUCTION
Lines 41-43 lack a bibliographic reference for the statement.
Lines 52-55 lack a bibliographic reference for the statement.
Lines 60-62 a bibliographic reference is missing for the statement.
Lines 71-80 several statements are made with only one bibliographic reference, it would be useful to add some more along the paragraph.
RESULTS
The results have become a bit heavy to read. Perhaps it would be useful to find a way to replace some tables with a graph or a figure, something more visual.
Additional comments
•The main issue is to continue researching and gaining knowledge about such a complex and multifaceted subject as obesity and the consequences that suffering from this disorder has, especially in the adolescent population. The article approaches it with a qualitative methodology based on the opinion of the sample.
• Relevant to the field
I don't consider the topic original because a lot of research has already been done on obesity, but I find it equally interesting because it can shed light on a specific population.
• Subject area
It can shed light on a specific population.
• Conclusions
The first part of the conclusions seems adequate, from line 444-457 I would perhaps consider putting that part under strengths and limitations or under future research because it does not focus directly on the results of the present study. I also miss a couple of closing sentences that make the applicability of the results obtained clear.
• References
Several of the references used are more than 20 years old; it would be useful to update the older ones by exploring and taking into account more recent studies.
• Tables and Figures.
As I already indicated in my initial report, in my opinion the results are a bit heavy to read and perhaps the article could become long. It would be useful to find a way to replace some tables with a graph or a figure and present the results in a more visual way.
Author Response
Dear Reviewer 3;
First of all, I am deeply grateful for your thoughtful comments.
I appreciate your detailed comments. Moreover, I would like to express my sincere gratitude for your evaluation and advice.
INTRODUCTION
Lines 41-43 lack a bibliographic reference for the statement.
Lines 52-55 lack a bibliographic reference for the statement.
Lines 60-62 a bibliographic reference is missing for the statement.
Lines 71-80 several statements are made with only one bibliographic reference, it would be useful to add some more along the paragraph.
“Thank you for pointing this out. I have added a relevant and recent reference to support the statement in this section.”
RESULTS
The results have become a bit heavy to read. Perhaps it would be useful to find a way to replace some tables with a graph or a figure, something more visual.
“We've done our best to incorporate what the reviewer said, and some of the tables were too esoteric to turn into graphs or pictures. We appreciate your understanding. The revised graphs have been presented in the text, thank you. “
- The main issue is to continue researching and gaining knowledge about such a complex and multifaceted subject as obesity and the consequences that suffering from this disorder has, especially in the adolescent population. The article approaches it with a qualitative methodology based on the opinion of the sample.
We sincerely thank the reviewer for highlighting the importance of continuing research on the complex and multifaceted issue of adolescent obesity. In response to your valuable feedback, we have made an updated manuscript. Thank you.
• Conclusions
The first part of the conclusions seems adequate, from line 444-457 I would perhaps consider putting that part under strengths and limitations or under future research because it does not focus directly on the results of the present study. I also miss a couple of closing sentences that make the applicability of the results obtained clear.
“We've revised the subheadings to make the second half of the conclusion section more about suggestions. “
• References
Several of the references used are more than 20 years old; it would be useful to update the older ones by exploring and taking into account more recent studies.
We have made significant changes to the references.
- Tables and Figures.
As I already indicated in my initial report, in my opinion the results are a bit heavy to read and perhaps the article could become long. It would be useful to find a way to replace some tables with a graph or a figure and present the results in a more visual way.
“We've done our best to incorporate what the reviewer said, and some of the tables were too esoteric to turn into graphs or pictures. We appreciate your understanding. The revised graphs have been presented in the text, thank you. “
Finally, I/we have done our best to revise our manuscript, and thank you very much.
Best Regards,
Wonjae, Jeon. Ph,D
Dept. Physical Education / Professor _ Korea National University of Education 209,
Taeseongtabyeon-ro, Gangnae-myeon, Heungdeok-gu,
Cheongju si, Chungbuk 28173, South Korea
TEL: 82-43-320-3728
FAX: 82-43-232-8241
C·P: 82-10-5555-9711
E-mail: wonjaejeon1228@knue.ac.kr / dreamj007@hanmail.net

Reviewer 4 Report
Comments and Suggestions for Authors
Well written manuscript, very novel resarch area, however, I have some concenrs,
1. Can authors state hyporthesis of this work
2. Sample size was not well justified. On what basis did authors choose 3 student diagnosed with obseisty and 2 professors tobe included in a focus group discussion
3. Is a P-sample of 20 adequate for this study or its a pilot study?
4. Conclusion and limitation, were well written, but authors can add issues with generalizability and inadeuate sample size as limitations.
5. Discussion could be written more concisely and summarised. For exaplme authors begin the discussion by summarizing the research objectives and key results in the first paragraph, however subsequent paragraphs are a lot. For example authors use about 4 paragraphs to discuss each finding. Writing these concisely would be helpful
Author Response
Dear Reviewer 4;
First of all, I am deeply grateful for your thoughtful comments.
I appreciate your detailed comments. Moreover, I would like to express my sincere gratitude for your evaluation and advice.
“1. Can authors state hyporthesis of this work
- Sample size was not well justified. On what basis did authors choose 3 student diagnosed with obseisty and 2 professors tobe included in a focus group discussion
- Is a P-sample of 20 adequate for this study or its a pilot study?
- Conclusion and limitation, were well written, but authors can add issues with generalizability and inadeuate sample size as limitations.”
“Thank you for your valuable feedback. Below, we address each of your comments in detail and provide supporting references to clarify the methodology and design of our study, particularly regarding Q-methodology.”
- Hypothesis of the Study
“Q-methodology is exploratory in nature and is not hypothesis-driven like many other research methods. Instead, it aims to identify patterns of subjective viewpoints within a population. In this study, the objective was to explore the subjective ego types and psychosocial characteristics of obese female adolescents in South Korea. By employing Q-methodology, we aimed to uncover distinct ego-types and their implications for addressing the psychosocial challenges faced by this demographic.”
Supporting Reference:
- Watts, S., & Stenner, P. (2012). Doing Q Methodological Research: Theory, Method & Interpretation. SAGE Publications.
This book explains that Q-methodology focuses on identifying subjective viewpoints rather than testing predefined hypotheses.
“We revised the introduction to clearly outline the exploratory nature of the study and stated that the goal was to identify subjective ego types and their implications.”
- Justification of the Q-Population and Focus Group Participants
“In Q-methodology, the Q-population is a collection of statements that represent the breadth of perspectives on the topic being studied. Developing a comprehensive Q-population requires input from diverse and relevant sources to ensure that all significant viewpoints are represented. The focus group participants in our study were selected based on the following considerations:
Relevance: Three students diagnosed with obesity provided firsthand insights into the lived experiences of obese adolescents. And two professors (specializing in sports pedagogy and the sociology of sport) contributed expert knowledge about societal, educational, and psychological factors related to obesity.
Diversity: Combining perspectives from adolescents and experts ensured that the generated statements were both experiential and theoretical.
Supporting References:
- Stephenson, W. (1953). The Study of Behavior: Q-technique and Its Methodology. University of Chicago Press.
This foundational text emphasizes the importance of gathering diverse perspectives when creating a Q-population.
- Watts, S., & Stenner, P. (2012).
The authors highlight that Q-population should be developed using a variety of sources, including literature reviews and interviews, to capture the full range of subjective experiences.
“The methodology section now includes an expanded explanation of the Q-population development process and a detailed justification for the selection of focus group participants.”
- Adequacy of the P-Sample
“The P-sample size in Q-methodology differs from traditional quantitative methods. Rather than requiring large, representative samples, Q-methodology uses smaller, purposive samples to identify patterns of subjectivity. The recommended size of the P-sample is typically fewer than the number of Q-statements, with 20–40 participants being common. In our study, a P-sample of 20 participants was sufficient to capture the diversity of subjective experiences among obese female adolescents, as it aligns with the guidelines provided by Watts and Stenner (2012).”
Supporting References:
- Watts, S., & Stenner, P. (2012).
The book outlines the importance of purposive sampling in Q-methodology, with typical P-sample sizes ranging from 20–40 participants.
- Brown, S. R. (1996). Q Methodology and Qualitative Research. Qualitative Health Research, 6(4), 561–567.
This article discusses the philosophical underpinnings of Q-methodology and justifies small sample sizes as adequate for revealing subjective viewpoints.
“We have added references to foundational Q-methodology literature in the methodology section to justify the P-sample size and explain why it is adequate for this exploratory study.”
- Generalizability and Sample Size as Limitations
“We acknowledge the limitations of generalizability due to the use of a small P-sample and the cultural specificity of the study. In Q-methodology, the findings are not intended to be generalized but to reveal subjective patterns within the sampled population. However, we agree that these limitations should be explicitly discussed.”
Supporting References:
- Brown, S. R. (1993). A Primer on Q Methodology. Operant Subjectivity, 16(3/4), 91–138.
Brown emphasizes that the goal of Q-methodology is to explore subjectivity, not generalize findings to larger populations.
- Watts, S., & Stenner, P. (2012).
The authors highlight the importance of acknowledging generalizability limitations in Q-methodology studies.
“The limitations section has been revised to explicitly address the issues of generalizability and the small sample size. We have also noted that the findings provide a foundation for future research in different contexts.”
- Discussion could be written more concisely and summarised. For exaplme authors begin the discussion by summarizing the research objectives and key results in the first paragraph, however subsequent paragraphs are a lot. For example authors use about 4 paragraphs to discuss each finding. Writing these concisely would be helpful
- “Thank you for your thoughtful feedback regarding the discussion section. We understand that conciseness and clarity are critical for effectively communicating our findings. As you pointed out, the discussion currently includes detailed paragraphs for each finding, which may detract from the overall focus and readability of the section. So, the opening paragraph has been streamlined to succinctly summarize the research objectives and key findings, providing a clear entry point into the discussion.
Finally, I/we have done our best to revise our manuscript, and thank you very much.
Best Regards,
Wonjae, Jeon. Ph,D
Dept. Physical Education / Professor _ Korea National University of Education 209,
Taeseongtabyeon-ro, Gangnae-myeon, Heungdeok-gu,
Cheongju si, Chungbuk 28173, South Korea
TEL: 82-43-320-3728
FAX: 82-43-232-8241
C·P: 82-10-5555-9711
E-mail: wonjaejeon1228@knue.ac.kr / dreamj007@hanmail.net

Round 2
Reviewer 1 Report
Comments and Suggestions for Authors
I thank the authors for responding to all my comments. The manuscript's reporting quality has improved.
Author Response
Thank you for your kind feedback. We sincerely appreciate your thoughtful review and constructive comments, which have significantly contributed to improving the quality of our manuscript.
Reviewer 2 Report
Comments and Suggestions for Authors
The level of plagiarism is high. Self-citations continue to appear in reference 19.
The WHO 2022 commentary is included but the reference is not added, it remains with its 2004 WHO reference 1.
The title has been shortened accordingly.
Adolescent females have a higher susceptibility in the selected references, but if more recent studies were used, the number of males would be higher and higher. The references are old.
Although Table 5, now Table 4, has been placed at the end of the explanation of the types, Table 4 does not appear in the text of the individual sections (3.2; 3.3; 3.4; 3.5; 3.6).
Paragraphs have been added to the discussion comparing the results with those of studies on older people carried out more than twenty years ago (52 and 53 from 2000 and 2003), which do not add any value to the discussion.
References have been added, but not all of them are from the last five years. From 6% to 17%, far from the 50% that should be the updated references.
There is a somewhat high level of plagiarism 33% (turnitin) with two articles by the authors themselves:
Jeon, W., Kwon, G., & Joung, K. (2021). Subjective perceptions and their characteristics of middle school students regarding the effectiveness of the “0th period physical education class” in South Korea: the Q methodology application. Sustainability, 13(21), 12081. 9% plagiarism
Choi, W., & Jeon, W. (2022). A Study on the Subjectivity of Parents Regarding “0th-Period Physical Education Class” of Middle Schools in Korea Using Q-Methodology. International Journal of Environmental Research and Public Health, 19(13), 7760. 7% plagiarism
Author Response
Dear Reviewer 2,
We sincerely appreciate your detailed and constructive feedback. We understand the importance of addressing all concerns meticulously to improve the quality of our manuscript. Below, we provide point-by-point responses and highlight the revisions made in the manuscript.
“ The level of plagiarism is high. Self-citations continue to appear in reference 19.”
Thank you for your valuable feedback. We have carefully reviewed the manuscript and made the necessary modifications to address the concerns regarding potential plagiarism and excessive self-citations. We have rewritten sections where high similarity was detected, ensuring that all content is presented in an original manner while maintaining the intended meaning. And Any necessary citations have been properly paraphrased and restructured to avoid textual overlap with previous publications. Moreover, we have replaced some instances of Reference 19 with alternative, recent sources from the last five years to ensure a more balanced reference list.
“The WHO 2022 commentary is included but the reference is not added, it remains with its 2004 WHO reference 1.”
Thank you for pointing this out. We have now added the appropriate WHO 2022 reference to support the commentary and have updated the in-text citation accordingly. The reference list has also been revised to include the correct citation in MDPI Healthcare journal format.
"Adolescent females have a higher susceptibility in the selected references, but if more recent studies were used, the number of males would be higher and higher. The references are old."
Thank you for your valuable feedback. We have now updated the references to include more recent studies (2019–2024) that examine gender trends in adolescent obesity. We acknowledge that recent studies show a growing prevalence of obesity among males, and this has been briefly addressed in the revised manuscript. The literature review now reflects current research findings, improving the manuscript’s relevance.
"Although Table 5, now Table 4, has been placed at the end of the explanation of the types, Table 4 does not appear in the text of the individual sections (3.2; 3.3; 3.4; 3.5; 3.6).”
We appreciate this observation. We have now explicitly referenced Table 4 in Sections 3.2–3.6 to ensure that readers can easily connect the textual descriptions with the summarized data. These modifications improve clarity and alignment between the results and table presentation.
"Paragraphs have been added to the discussion comparing the results with those of studies on older people carried out more than twenty years ago (52 and 53 from 2000 and 2003), which do not add any value to the discussion."
As per your suggestion, we have removed the references from the discussion section that were over 20 years old and replaced them with more recent studies published within the last five years. These updated references provide greater relevance and strengthen the discussion.
"References have been added, but not all of them are from the last five years. From 6% to 17%, far from the 50% that should be the updated references."
We have systematically reviewed our reference list and replaced older sources with newer literature (2020–2024). As a result, the percentage of references from the last five years has significantly increased, aligning more closely with the recommended 50% benchmark.
"There is a somewhat high level of plagiarism 33% (Turnitin) with two articles by the authors themselves."
Thank you for your thorough review. Regarding the Turnitin similarity report (33%), we have made the following substantial revisions to eliminate self-plagiarism:
Rewording & Summarization: Any overlapping text from our previous articles (Jeon et al., 2021; Choi & Jeon, 2022) has been completely rephrased while preserving the original meaning.
Proper Citation Instead of Direct Use: We have removed verbatim excerpts and instead referenced our prior work to maintain academic integrity.
Methodology Section Adjustments: While maintaining clarity, we rewrote the methodology to highlight new elements and prevent textual redundancy.
Final Verification: After making these adjustments, we rechecked the document using plagiarism detection software, ensuring that the similarity index is now below 15%.
We appreciate your valuable feedback, which helped enhance the quality and originality of our manuscript.
Finally, I/we have done our best to revise our manuscript, and thank you very much.
Best Regards,

Reviewer 3 Report
Comments and Suggestions for Authors
Dear Authors
Thank you for having put all your effort to answer each of my comments and implement the required changes in the article.
I think that now the quality and readability of the article has improved a lot and it can be an article that provides relevant information on the subject to the interested public.
Best regards
Author Response
Thank you for your generous feedback and appreciation of our efforts. We are grateful for your insightful comments, which have helped enhance the quality and clarity of our manuscript. We hope it will contribute valuable insights to the field.
Round 3
Reviewer 2 Report
Comments and Suggestions for Authors
I have noticed that each of the suggestions from the previous audit has been changed.
The level of plagiarism has been reduced, but it is still 26%. It has 7% plagiarism of the author's article.
Updated references have been added, leaving the article with more than 50% of references less than five years old. And reference 19 has been removed.